# Self-reported uptake of STI testing services among adolescents and young people aged 15–24 years: Findings from the Yathu Yathu cluster randomized trial in Lusaka, Zambia

**Bernadette Hensen**[1]*, **Mwelwa M. Phiri**[2], **Lucheka Sigande**[2], **Ab Schaap**[2,3], **Melvin Simuyaba**[2], **Rosemary Zulu-Phiri**[2], **Louis Mwape**[2], **Sian Floyd**[3], **Sarah Fidler**[4], **Richard Hayes**[3], **Musonda Simwinga**[2], **Helen Ayles**[2,5]

1 Department of Public Health, Institute of Tropical Medicine, Antwerp, Belgium, 2 Zambart, Lusaka, Zambia, 3 Department of Infectious Disease Epidemiology, London School of Hygiene and Tropical Medicine, London, United Kingdom, 4 Department of Infectious Disease, Imperial College, London, United Kingdom, 5 Department of Clinical Research, London School of Hygiene and Tropical Medicine, London, United Kingdom

* bhensen@itg.be

## Abstract

There is little evidence regarding community-based delivery of STI testing and treatment for youth aged 15–24 (AYP) in Zambia. In a cluster-randomised trial, we evaluated whether offering syndromic STI screening through community-based, peer-led sexual and reproductive health services (Yathu Yathu) with referral to a local health facility for testing, increased self-reported testing for STIs (other than HIV) among AYP. Two communities in Lusaka were divided into 10 zones each (20 zones in total); by community, zones were randomly allocated (1:1) to Yathu Yathu or control. Monitoring data were used to describe syndromic STI screening through Yathu Yathu and an endline cross-sectional survey used to evaluate the impact of Yathu Yathu on self-reported ever and recent (last 12 months) STI testing. 10,974 AYP accessed Yathu Yathu; 66.6% (females—67.7%; males—64.7%) were screened for STIs, 6.2% reported any STI symptoms. In the endline survey, 23.3% (n = 350/1501) of AYP who ever had sex ever STI tested; 13.5% (n = 174/1498) who had sex in the last 12 months recently STI tested. By trial arm, there was no difference in self-reported ever or recent STI testing among all AYP. Among men aged 20–24, there was evidence that ever STI testing was higher in the Yathu Yathu compared to control arm (24.1% vs 16.1%; adjPR = 1.67 95%CI = 1.02, 2.74; p = 0.04). Among AYP who ever STI tested, 6.6% (n = 23) reported ever being diagnosed with an STI. Syndromic STI management through community-based, peer-led services showed no impact on self-reported STI testing among AYP. Research on community-based delivery of (near) point-of-care diagnostics is needed.

**Trial registration number(s)**: NCT04060420 https://clinicaltrials.gov/ct2/show/NCT04060420; and ISRCTN75609016; https://doi.org/10.1186/ISRCTN75609016.

**Data Availability Statement:** Location of the data: LSHTM Data Compass: https://doi.org/10.17037/DATA.00003742.

**Funding:** This research was jointly funded by the UK Medical Research Council (MRC) and the Foreign Commonwealth and Development Office (FCDO) under the MRC/FCDO Concordat agreement, together with the Department of Health and Social Care (DHSC) (grant number MR/R022216/1 to HA). The funders had no role in the study design, data collection and analysis, decision to publish, or preparation of the manuscript.

**Competing interests:** The authors have declared that no competing interests exist.

## Introduction

In 2020, there were an estimated 374 million new sexually transmitted infections (STIs) globally with one of four curable STIs, namely chlamydia, gonorrhoea, syphilis and trichomoniasis [1]. The burden of STIs is particularly high among adolescents and young people aged 10–24, with an estimated incidence of 18.4 per 100-person years for chlamydia, genital herpes, gonorrhoea, syphilis, and trichomoniasis [2]. Despite a high incidence of STIs, including HIV, STI control strategies are failing to reach youth [3]. A recent trend analysis for the Global Burden of Diseases, Injuries, and Risk Factors Study found that, although STI incidence decreased among adolescents and young people aged 10 to 24 between 2010 and 2019, incidence increased among adolescents aged 10 to 14, from 1158.9 per 100,000 population in 1990 to 1215.4 per 100,000 population in 2019 [2]. The study also found that adolescents and young people in countries in southern Africa had the highest incidence of STIs, including HIV, in 2019 [2].

In many countries, particularly countries with limited diagnostic resources, STI diagnosis and treatment relies on syndromic management. Recommended by the World Health Organization (WHO) in the 1990s [4], it is now clear that syndromic management is inadequate: many STIs are asymptomatic and many STI symptoms are nonspecific, particularly among adolescent girls and women [5]. Syndromic management leads to overtreatment of symptoms not associated with STIs, posing a risk for antimicrobial resistance, and undertreatment of asymptomatic STIs, a risk for ongoing transmission and morbidity [6]. As cheaper point-of-care (or near point-of-care) tests become available [7], including dual syphilis/HIV self-tests [8], evidence of effective strategies to deliver these tests to adolescents and young people (AYP) will be needed. A key gap, however, in designing effective STI control strategies is the lack of country-specific data on the burden of STIs among AYP, particularly from southern African countries [9].

In Zambia, there are few recent data on the prevalence of STIs among AYP; recent available evidence includes a study conducted between 2016–2019 on the prevalence of chlamydia and gonorrhoea among female sex workers recruited through community outreach and single mothers with children aged <5 recruited through post-natal clinics [10]. Studies conducted more than a decade ago have reported: the prevalence of chlamydia and gonorrhoea among women living with HIV in Lusaka [11]; the prevalence of genital tract infections, including trichomoniasis and gonorrhoea, among pregnant women living with HIV in Lusaka [12], and the epidemiology of chlamydia, gonorrhoea, and syphilis among adolescents and adults aged 15–49 in four African cities, including Lusaka [13].

Recognizing the need to reach AYP with sexual and reproductive (SRH) services, we evaluated the impact of Yathu Yathu—comprehensive, community-based, and peer-led SRH services–on knowledge of HIV status [14]. Among the services available through the Yathu Yathu community-based spaces (called hubs) was symptomatic screening of STIs, other than HIV, with referral to the local health facility for testing and treatment. Using a cluster randomized trial (CRT) design, we found that the intervention increased knowledge of HIV status among AYP [15]. In this paper, we use data routinely collected during implementation of Yathu Yathu to describe symptomatic STI screening among AYP accessing Yathu Yathu. We also evaluate the impact of Yathu Yathu on self-reported testing for STIs, other than HIV, describe the self-reported prevalence of STIs and factors associated with ever STI-testing.

## Methods

We report analyses of the CRT in line with the CONSORT extension for Cluster Randomized Trials (S1 Checklist) [16].

## Study location and population

The Yathu Yathu CRT was conducted in two high-density, urban communities in Lusaka, Zambia. As described elsewhere, the two communities were divided into 20 zones of roughly equal population size and, within each community, zones were randomly allocated to the Yathu Yathu intervention or control group [14]. Immediately following randomization, in August 2019, a census was conducted in all 20 zones. All households and all AYP aged 15–24 resident in these households were enumerated. AYP aged 15–24 were offered a prevention points card (PPC) and informed that they could gain points for services accessed at a Yathu Yathu hub or health facility (intervention arm) or at the health facility only (control arm), and could use these points to "buy" rewards (soap, toothbrush, toothpaste, deodorant). The PPC served to incentivize service access and allowed the study to monitor service use, inform adaptations [17], measure implementation and evaluate the impact of Yathu Yathu on coverage of key SRH services [14].

## Yathu Yathu intervention

The Yathu Yathu intervention, described elsewhere [14], consisted of two main components: 1) community-based delivery of comprehensive SRH services through hubs that were managed by peer support workers and 2) the PPC, linked to rewards to incentivize use of services. The points AYP gained when accessing services were dependent on the psychological difficulty of accessing each individual service, as determined by AYP during formative research to design the intervention [18]. Community engagement was conducted throughout implementation.

The ten hubs, one in each intervention zone, were managed day-to-day by peer support workers, who were supported by a hub supervisor, and a nurse who visited each hub weekly. The services available on-site included, but were not limited to, HIV testing (finger-prick and oral self-testing), symptomatic screening for STIs, comprehensive sexuality education, and contraceptives (oral pill and injectables; condoms). AYP could accrue 125 points for symptomatic STI screening (similar to collection and return of an oral HIV self-testing kit and TB screening); for AYP experiencing symptoms, peer support workers referred the individual to the local health facility for STI diagnosis and treatment. If AYP were referred to the health facility, tested positive for an STI and initiated treatment for this STI, they could gain 250 points for initiating treatment (either through informing a peer support worker at the hub or information desk at the local health facility). Yathu Yathu services were available from September 2019.

Each community had one public health facility offering STI testing and treatment services and youth-friendly corners; as such, these were the services available in the control arm. An information desk was established at both facilities to welcome AYP and allow them to accrue points on their PPC for services accessed at the health facility.

## Data collection

The first data source was process data on services accessed by AYP attending Yathu Yathu hubs, which were routinely collected using the PPC. Using these data, we described (overall and by sex): the number and percentage of AYP screened for STIs; the percentage of AYP experiencing any of the following symptoms: pain during sex, genital itching, discharge and/or sores, and the percentage of AYP with any symptoms who were referred for STI testing. Data on services accessed were collected from 1 September 2019, when services were first available, until 30 September 2021.

The second data source was the endline population-based survey used to evaluate the impact of Yathu Yathu on the trial outcomes [15]. The sampling frame for the survey was AYP who accepted a PPC during the census. In line with the sample size calculation for the primary outcome, approximately 2000 AYP participated in the endline survey. This survey was initiated on 29 April 2021 and completed 4 November 2021. The questionnaire administered during the survey included modules on socio-demographics, sexual behavior, uptake of HIV testing services and linkage to prevention or care, and whether AYP had ever and recently (last 12 months) tested for an STI, other than HIV, and whether they ever tested positive for an STI.

## Outcomes and explanatory variables

In the endline survey, the two outcomes of interest were: the percentage of AYP self-reporting ever testing for an STI, other than HIV, among AYP reporting ever having had sex, and the percentage of AYP self-reporting recently (last 12 months) testing for an STI among AYP reporting sex in the last 12 months. Variables explored for their association with ever STI testing included: age, sex, highest level of education attained, employment status, whether AYP reported ever having no food to eat in their household in the last 4 weeks because of a lack of resources to get food, and sexual behaviors, including number of lifetime sexual partners, number of sexual partners in the last 12 months, and condom use at last sex.

## Data analysis

To estimate whether Yathu Yathu had an impact on self-reported ever and recent STI testing, we used the two-stage process recommended for CRTs with <15 clusters (zones)/arm [19]. We first estimated the percentage of AYP self-reporting each outcome by cluster, then estimated the average of these cluster-specific estimates by arm. To formally compare the two trial arms in an "unadjusted" analysis, we fitted a linear regression model of log(cluster-level proportion) on trial arm and community, to obtain a log prevalence ratio (PR) comparing the Yathu Yathu arm with the control arm and a corresponding 95% confidence interval. To adjust for age and sex, we used a two stage process. First, we fitted a logistic regression model to individual-level data for both outcomes, ignoring allocation to trial arm, but adjusting for age, sex, and community. We then estimated an individual's predicted probability of the outcomes after adjusting for these covariates. Subsequently, we aggregated the observed (O) and expected (E) numbers of individuals with the outcomes, under the null hypothesis of no intervention effect. We then estimated the ratio of the O/E number of individuals self-reporting ever and recently STI testing, and calculated the log(ratio-residual) as log(O/E). Next, to obtain a PR and associated 95% confidence interval comparing the Yathu Yathu and control trial arms adjusted for sex, age, and community, we fitted a linear regression model of log(O/E) on trial arm and community. We stratified analyses by four age-sex groups (adolescent girls and boys aged 15–19, young women and men aged 20–24).

To explore factors associated with ever STI testing, we used logistic regression with robust standard errors to account for clustering by the 20 zones. As COVID-19 control measures likely affected STI service delivery at the health facility in the year prior to the survey, we did not explore factors associated with recent STI-testing. We conducted analyses separately for adolescent girls and young women (AGYW) and adolescent boys and young men (ABYM), as we expected factors associated with uptake might differ by sex.

Factors found to be associated with ever STI-testing in age- and community-adjusted analyses at the p<0.1 level were included in multivariable analyses. In these analyses, we used a hierarchical approach to avoid over-adjusting for variables likely to be mediators of a relationship. For example, to explore whether household-level wealth and food availability

were associated with STI testing we did not adjust for individual-level factors, which are likely to be on the causal pathway between household-level variables and self-reported uptake of STI testing.

## Ethical statement

The Yathu Yathu CRT was approved by the University of Zambia Biomedical Research Ethics Committee (ref 007-04-19) and the London School of Hygiene and Tropical Medicine (ref 17104). All AYP were asked for written informed consent prior to PPC distribution and to participate in the endline survey. For the survey, a waiver of parental consent for AYP aged <18-years was granted by both institutional reviews boards, as parents/guardians provided consent during census and PPC distribution for adolescents aged 15 to 17-years. The study was approved by the Zambian National Health Research Authority. This trial was registered at: NCT04060420 https://clinicaltrials.gov/ct2/show/NCT04060420; and ISRCTN75609016 https://doi.org/10.1186/ISRCTN75609016.

## Results

### Syndromic STI screening at Yathu Yathu services

Among the 10,974 AYP who accessed any Yathu Yathu service, 66.6% (67.7%, n = 4743/7009, AGYW; 64.7%, n = 2567/3965, ABYM) of AYP were screened for an STI at least once. Among these AYP, 39.6% (n = 2898/7310) were screened more than once and 6.2% (n = 679) reported any STI symptoms at any visit (10.8%, n = 511 AGYW, and 6.5%, n = 168 ABYM).

At their first STI symptom screen, 7.7% (n = 563/7310) of AYP reported experiencing any STI symptom (8.8%, n = 419, of AGYW, and 5.6%, n = 144, of ABYM; Fig 1A); all were referred for STI testing. The most common symptom, among AYP experiencing any symptoms, was genital itching (75.3%, n = 424) followed by pain during sex (30.2%, n = 170), discharge (29.0%, n = 163) and genital sores (27.7%, n = 156). Reported symptoms were similar by sex (Fig 1B), however, there was some evidence that AGYW were more likely to report genital discharge (30.8% vs 23.1%, respectively).

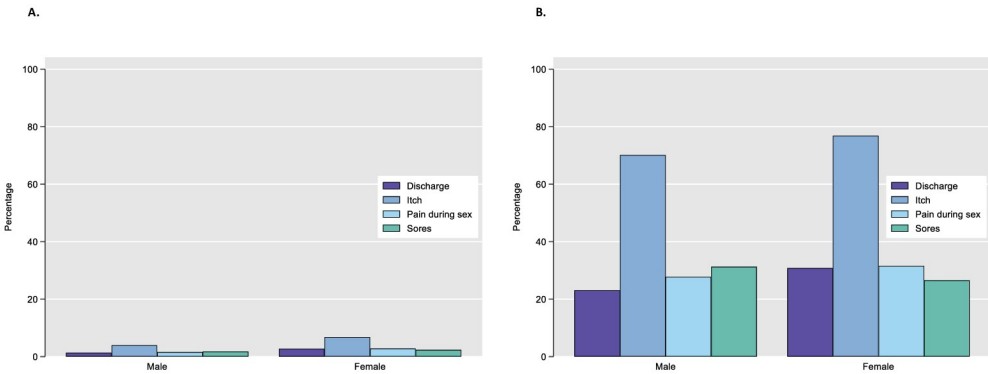

**Fig 1. Self-reported STI symptoms among all adolescents and young people aged 15–24 screened for an STI (A, N = 7310) and self-reporting any STI symptoms at their first symptomatic screen for STIs through Yathu Yathu (B, N = 563), 2019–2021.**

## Self-reported ever and recent STI testing in the endline survey

Overall, 1989 AYP participated in the endline survey, among whom 75.7% (n = 1505) reported ever having had sex and 1501 had complete data on testing for STIs (Fig 2).

Among these AYP, just over half were female (52.2%; n = 784/1501) and 59.9% (n = 899/1501) were aged 20–24 years at the time of the census (Table 1). Almost half reported attaining incomplete secondary education (48.4%, n = 727/1501) and one-third were currently employed (34.6%, n = 519). Approximately half (44.8%; n = 673) reported 2–4 lifetime sexual partners, 73.5% (n = 1101) had sex in the last 12-months, and 45.2% (n = 677) reported using a condom the last time they had sex (Table 1).

By trial arm, there was no difference in self-reported ever (24.1%, n = 182/754, vs 22.5%, n = 168/747, respectively, adjPR = 1.16 95%CI 0.70, 1.91; p = 0.56) or recent (15.1%, n = 86/564, vs 11.1%, n = 62/535, respectively, adjPR = 1.70 95%CI 0.88, 3.27; p = 0.11) STI testing among all AYP (Table 2). However, there was evidence that ever STI testing was higher among young men aged 20–24 in the Yathu Yathu compared to control arm (24.1%, n = 55/227, vs 16.1%, n = 35/212, respectively, adjPR = 1.67 95%CI = 1.02, 2.74; p = 0.04).

Among AYP who ever tested for an STI, 6.6% (n = 23/350) reported ever being diagnosed with an STI; five individuals did not know which STI they tested positively for. Among the remaining 18 individuals, 83.3% (n = 15) self-reported being diagnosed with syphilis and 16.7% (n = 3) with gonorrhoea. Overall, among AYP reporting ever-testing for an STI, the self-reported prevalence of ever-testing positive for syphilis was 4.3% (n = 15/350) and for gonorrhoea was 0.9% (n = 3/350).

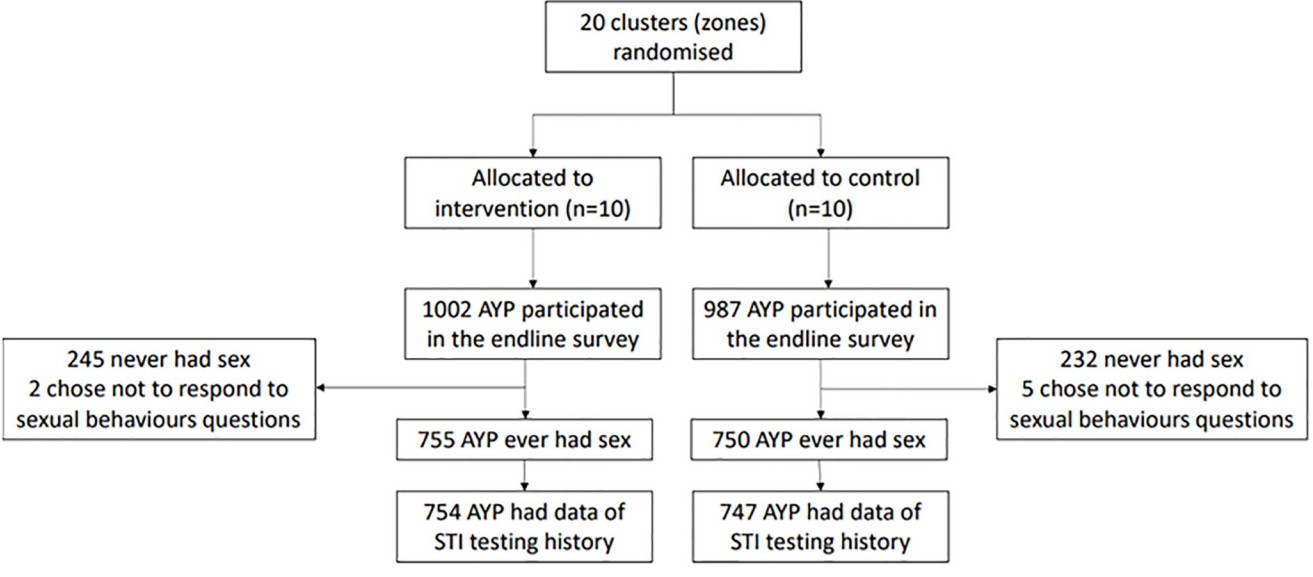

AYP – Adolescent and young person aged 15-24

**Fig 2.**

**Table 1. Characteristics of AYP who consented to participate in the endline survey and reported ever having had sex, 2021.**

| Overall | 1501 |
|---|---|
| **Sex** | |
| Male | 717 (47.8) |
| Female | 784 (52.2) |
| **Age group (at time of census)** | |
| 15-19y | 602 (40.1) |
| 20-24y | 899 (59.9) |
| **Current marital status** | |
| Single–never married | 1111 (74.0) |
| Currently married &/or living with partner | 350 (23.3) |
| Currently married, living apart | 25 (1.7) |
| Divorced, separated or widowed | 15 (1.0) |
| **Currently in school** | |
| No | 1204 (80.0) |
| Yes | 301 (20.0) |
| **Highest level of education attained** | |
| None/incomplete primary | 63 (4.2) |
| Complete secondary | 137 (9.1) |
| Incomplete secondary | 727 (48.4) |
| Complete secondary/Higher | 574 (38.2) |
| **Currently employed** | |
| No | 982 (65.4) |
| Yes | 519 (34.6) |
| **In past 4 weeks, ever no food to eat in the household because of lack of resources** | |
| No | 965 (64.3) |
| Yes | 536 (35.7) |
| **Wealth quintiles according to household assets** | |
| 1 –Lowest | 317 (21.2) |
| 2 | 344 (23.0) |
| 3 –Middle | 276 (18.4) |
| 4 | 268 (17.9) |
| 5—Highest | 293 (19.6) |
| **Lifetime number of sex partners** | |
| 1 | 577 (38.4) |
| 2 to 4 | 673 (44.8) |
| 5 to 10 | 184 (12.3) |
| >10 | 67 (4.5) |
| **Condom used at last sex** | |
| No | 822 (54.8) |
| Yes | 677 (45.2) |
| **Had sex in the last 12 months** | |
| No | 398 (26.6) |
| Yes | 1101 (73.4) |
| **Number of sex partners in the past 12 months*** | |
| 1 | 859 (78.0) |
| 2 to 4 | 198 (18.0) |

(*Continued*)

**Table 1.** (Continued)

| Overall | 1501 |
|---|---|
| >5 | 44 (4.0) |

*Among those who reported sex in the last 12 months (N = 1101).

Overall, 23.3% (n = 350/1501) of AYP who ever had sex reported ever testing for an STI, other than HIV. Restricting to AYP who reported ever having sex in the last 12 months, 13.5% (n = 148/1099) reported testing for an STI in the last 12 months.

## Factors associated with self-report of ever-testing for STIs

Among AGYW who ever had sex, 58.7% (n = 462) were aged 20–24 at the time of the census and 58.5% (n = 460) were currently single and had never been married. Thirty-percent (31.5%, n = 248) had completed secondary education or higher, and the majority were not currently employed (79.0%; n = 622). Over half (59.6%, n = 465) had ever given birth and 46.5% (n = 366) had 2–4 lifetime sexual partners. Almost all reported one sexual partner in the last 12 months (90.9%, n = 562) and 37.9% (n = 298) used a condom at last sex.

**Table 2. Self-reported ever- and recent-testing for an STI, other than HIV, among adolescents and young people, by trial arm, 2021.**

| Ever STI-tested (among AYP reporting ever having had sex) | Yathu Yathu Arm | Control Arm | Adjusted PR[2] | 95% CI | p-value |
|---|---|---|---|---|---|
| **Overall** | 24.1%[3] (n = 182/754)[4] | 22.4%[5] (n = 168/747)[6] | 1.16 | 0.70, 1.91 | 0.56 |
| Adolescent girls (aged 15–19[1]) | 23.8% (n = 38/161) | 23.6% (n = 39/163) | 0.93 | 0.45, 1.92 | 0.85 |
| Adolescent boys (aged 15–19[1]) | 12.2% (n = 16/133) | 11.1% (n = 14/145) | 1.23 | 0.62, 2.46 | 0.53 |
| Women (aged 20–24[1]) | 31.8% (n = 73/233) | 34.3% (n = 80/227) | 0.91 | 0.52, 1.58 | 0.73 |
| Men (aged 20–24[1]) | **24.1%** **(n = 55/227)** | **16.1%** **(n = 35/212)** | **1.67** | **1.02, 2.74** | **0.04** |
| **Recently STI-tested (among AYP reporting sex in last 12 months)** | **Yathu Yathu Arm** | **Control Arm** | **Adjusted PR[2]** | **95% CI** | **p-value** |
| **Overall** | 15.1% (n = 86/564) | 11.1% (n = 62/535) | 1.70 | 0.88, 3.27 | 0.11 |
| Adolescent girls (aged 15–19[1]) | 14.5% (n = 17/115) | 14.5% (n = 16/117) | 1.04 | 0.52, 2.11 | 0.90 |
| Adolescent boys (aged 15–19[1]) | 11.2% (n = 8/89) | 3.3% (n = 2/72) | 1.26 | 0.73, 2.17 | 0.38 |
| Women (aged 20–24[1]) | 19.4% (n = 38/192) | 14.9% (n = 31/191) | 1.26 | 0.57, 2.76 | 0.54 |
| Men (aged 20–24[1]) | 14.7% (n = 24/166) | 8.6% (n = 12/157) | 1.74 | 0.87, 3.47 | 0.11 |

[1] Age at time of consent to receive a prevention points card,

[2] PR = Prevalence ratio. Overall: Adjusted for age, sex and community. Each age-sex group: Adjusted for community;

[3] Arithmetic mean of the 10 cluster-specific values of the proportion of AYP who reported ever/recently STI testing in the Yathu Yathu arm;

[4] n = number of individuals who ever/recently STI tested in the intervention arm, denominator = number of survey participants in the intervention arm who reported ever/recently having had sex;

[5] Arithmetic mean of the 10 cluster-specific values of the proportion of AYP who reported ever/recently STI testing in the control arm;

[6] n = number of individuals who ever/recently STI tested in the control arm, denominator = number of survey participants in the control arm who reported ever/recently having had sex.

Age was strongly associated with ever STI-testing, with 33.3% (n = 153/462) of young women aged 20–24 ever STI-testing compared to 23.8% (n = 77/325) of adolescent girls (adjOR = 1.60 95%CI 1.21, 2.12; p = 0.004; Table 3). Similarly, being currently married and/or living with a partner (41.7%, n = 131/315, vs never married 20.7%, n = 95/460; adjOR = 3.00 95%CI 2.06, 4.36; p<0.001) and ever giving birth (37.7% n = 175/465, vs never given birth 17.3%, n = 54/315; adjOR = 2.54 95%CI 1.70, 3.78; p<0.001) were strongly associated with ever STI-testing. Self-reported ever STI-testing was higher among AGYW who reported ever having no food in the household in the past 4-weeks (35.6%, n = 104/294, vs 25.6%, n = 126/493; adjOR = 1.39 95%CI 1.03, 1.88; p = 0.03). There was little evidence that sexual behaviors were associated with ever STI-testing among AGYW.

Sixty-one percent (61.3%, n = 439) of ABYM who reported ever having had sex were aged 20–24 at the time of the census; 91.0% (n = 653) were currently single and had never been married (Table 4). Forty-five percent (45.4%, n = 326) had completed secondary or higher education. Forty-three percent (43.0%, n = 308) reported 2–4 lifetime sexual partners, while 8.9% (n = 64) reported >10 lifetime sexual partners. Just over half reported condom use at last sex (53.0%, n = 380); among ABYM who reported sex in the last 12 months, over half (61.7%, n = 300) reported one sexual partner in the last 12 months.

Similar to AGYW, young men aged 20–24 were more likely to report ever STI-testing compared to adolescent boys (20.5%, n = 90/440 vs 10.8%, n = 30/278; adjOR = 2.13 95%CI 1.36, 3.32; p = 0.001). There was strong evidence that reporting more sex partners in the last 12 months (2 to 4: 25.9% n = 38/147 vs one: 12.3% n = 37/301; adjOR = 2.54 95%CI 1.59, 4.07; p<0.001) was associated with ever STI-testing. There was weak evidence of an association between higher levels of educational attainment (complete secondary/higher education: 21.5% n = 70/326 vs none/incomplete primary or complete primary education: 6.9% n = 5/73; adjOR = 3.35 95%CI 1.16, 9.66; p = 0.06) and ever-testing, and that ABYM residing in a household with higher relative wealth were more likely to report ever STI-testing than ABYM in lower wealth quintiles (highest (5): 22.8% n = 36/158 vs lowest (1): 14.2% 19/135; adjOR = 1.84 95%CI 0.98, 3.47; p = 0.07).

## Discussion

We found no evidence that symptomatic STI screening offered through comprehensive, community-based, peer-led SRH services had an impact on self-reported uptake of STI testing services. Only a quarter of AYP who ever had sex self-reported ever STI-testing, with only one in ten reporting testing for an STI, other than HIV, in the previous 12 months. By sex, there was evidence of an effect on self-reported ever STI-testing among young men aged 20–24. Ever STI testing was higher among AGYW; among whom, being married and having ever given birth were associated with ever STI testing. Higher educational attainment was associated with ever STI-testing among all AYP. Only one in five ABYM ever STI-tested; among ABYM, but not AGYW, ever STI testing was associated with sexual behaviors.

Despite evidence of an impact on knowledge of HIV status [15], we found no evidence that Yathu Yathu increased uptake of STI testing overall. Importantly, STI testing and treatment were only available at the local health facility. This finding may reflect inaccessibility of STI services at the local health facility, due to waiting times, distances to health facilities, and lack of services that are acceptable to youth, as reported in other studies [20]. We did, however, observe an effect on ever STI testing among young men aged 20–24. In our primary outcome analysis, we found that Yathu Yathu had the greatest effect on knowledge of HIV status among adolescent boys [15]. These combined findings among ABYM may reflect their limited engagement with the formal health system. Health facilities are often considered "female

**Table 3. Factors associated with ever STI-testing among adolescent girls and young women self-reporting ever having had sex in the Yathu Yathu endline survey, 2021 (N = 784).**

| | Characteristics (n, column %) and self-reported ever STI-tested (n, row%) | | Age- and community-adjusted OR (95%CI) | Final adjusted OR[1] | p-value |
|---|---|---|---|---|---|
| **Age group** | | | | | |
| 15-19y | 324 (41.3) | 77 (23.8) | 1 | 1 | 0.001 |
| 20-24y | 460 (58.7) | 153 (33.3) | 1.60 (1.21, 2.12) | 1.60 (1.21, 2.12) | |
| **Current marital status** | | | | | |
| Single–never married | 458 (58.4) | 95 (20.7) | 1 | 1 | <0.001 |
| Currently married &/or living with partner | 314 (40.1) | 131 (41.7) | 2.53 (1.73, 3.71) | 3.00 (2.06, 4.36) | |
| Currently married, living apart; Divorced, separated or widowed | 12 (1.5) | 4 (33.3) | 1.99 (0.49, 8.05) | 2.19 (0.57, 8.40) | |
| **Currently in school** | | | | | |
| No | 639 (81.5) | 198 (31.0) | 1 | 1 | 0.65 |
| Yes | 145 (18.5) | 32 (22.1) | 0.73 (0.47, 1.12) | 0.90 (0.58, 1.41) | |
| **Highest level of education attained** | | | | | |
| None/(in)complete primary | 127 (16.2) | 40 (31.5) | 1 | 1 | <0.001 |
| Incomplete secondary | 409 (52.2) | 109 (26.7) | 0.86 (0.61, 1.22) | 1.04 (0.71, 1.52) | |
| Complete secondary/Higher | 248 (31.6) | 81 (32.7) | 1.14 (0.73, 1.77) | 1.84 (1.14, 2.97) | |
| **Currently employed** | | | | | |
| No | 619 (79.0) | 190 (30.7) | 1 | 1 | 0.19 |
| Yes | 165 (21.0) | 40 (24.2) | 0.68 (0.46,1.01) | 0.74 (0.47, 1.16) | |
| **In past 4 weeks, ever no food to eat in the household because of lack of resources** | | | | | |
| No | 492 (62.8) | 126 (25.6) | 1 | 1 | 0.03 |
| Yes | 292 (37.2) | 104 (35.6) | 1.38 (1.03, 1.84) | 1.39 (1.03, 1.88) | |
| **Wealth quintiles according to household assets** | | | | | |
| 1—Lowest | 183 (23.3) | 63 (34.4) | 1 | 1 | 0.08 |
| 2 | 188 (24.0) | 52 (27.7) | 0.80 (0.53, 1.21) | 0.84 (0.55, 1.29) | |
| 3—Middle | 154 (19.6) | 35 (22.7) | 0.59 (0.39, 0.90) | 0.63 (0.41, 0.97) | |
| 4 | 124 (15.8) | 36 (29.0) | 0.86 (0.50, 1.46) | 0.96 (0.57, 1.62) | |
| 5—Highest | 135 (17.2) | 44 (32.6) | 0.92 (0.63, 1.35) | 1.05 (0.69, 1.58) | |
| **Ever given birth** | | | | | |
| No | 313 (40.3) | 54 (17.3) | 1 | 1 | <0.001 |
| Yes | 464 (59.7) | 175 (37.7) | 2.80 (1.84, 4.25) | 2.54 (1.70, 3.78) | |
| **Lifetime number of sex partners** | | | | | |
| 1 | 379 (48.3) | 106 (28.0) | 1 | 1 | 0.55 |
| 2 to 4 | 365 (46.6) | 111 (30.4) | 1.06 (0.77, 1.44) | 1.17 (0.87, 1.58) | |
| >5 | 40 (5.1) | 13 (32.5) | 1.00 (0.55, 1.84) | 1.07 (0.61, 1.89) | |
| **Condom used at last sex** | | | | | |
| No | 486 (62.1) | 153 (31.5) | 1 | 1 | 0.65 |
| Yes | 297 (37.9) | 77 (25.9) | 0.88 (0.56, 1.37) | 1.12 (0.70, 1.78) | |
| **Had sex in the last 12 months** | | | | | |
| No | 168 (21.5) | 29 (17.3) | 1 | 1 | 0.12 |
| Yes | 615 (78.5) | 200 (32.5) | 2.09 (1.33, 3.29) | 1.47 (0.91, 2.38) | |
| **Number of sex partners in the past 12 months** | | | | | |
| 1 | 559 (90.9) | 187 (33.5) | 1 | 1 | 0.09 |
| 2+ | 56 (9.1) | 13 (23.2) | 0.58 (0.30, 1.09) | 0.60 (0.33, 1.08) | |

OR = Odds ratio;

[1]. In adjusted analyses, further adjusted for marital status, educational attainment, currently employed, wealth quintile and availability of food in the household; with the exception of wealth quintile and availability of food in the household, which were not additionally adjusted for other variables (see Methods).

**Table 4. Factors associated with ever STI-testing among adolescent boys and young men self-reporting ever having had sex in the Yathu Yathu endline survey, 2021 (N = 717).**

| | Characteristics (n, column %) and self-reported ever STI testing (n, row%) | | Age- and community-adjusted OR (95%CI)[1] | Final adjusted OR[2] | p-value |
|---|---|---|---|---|---|
| **Age group** | | | | | |
| 15-19y | 278 (38.7) | 30 (10.8) | 1 | 1 | 0.001 |
| 20-24y | 439 (61.3) | 90 (20.5) | 2.13 (1.36, 3.32) | 2.13 (1.36, 3.32) | |
| **Current marital status** | | | | | |
| Single–never married | 653 (91.1) | 107 (16.4) | 1 | 1 | 0.49 |
| Currently married, living apart; Divorced, separated or widowed | 64 (8.9) | 13 (20.1) | 0.99 (0.47, 2.12) | 1.29 (0.62, 2.69) | |
| **Currently in school** | | | | | |
| No | 561 (78.2) | 99 (17.7) | 1 | 1 | 0.29 |
| Yes | 156 (21.8) | 21 (13.5) | 0.92 (0.61, 1.40) | 0.78 (0.49, 1.24) | |
| **Highest level of education attained** | | | | | |
| None/incomplete primary or Complete primary | 73 (10.2) | 5 (6.9) | 1 | 1 | 0.06 |
| Incomplete secondary | 318 (44.4) | 45 (14.2) | 2.52 (0.90, 7.11) | 2.55 (0.91, 7.10) | |
| Complete secondary/Higher | 326 (45.4) | 70 (21.5) | 3.52 (1.21, 10.21) | 3.35 (1.16, 9.66) | |
| **Currently employed** | | | | | |
| **No** | 363 (50.6) | 54 (14.9) | 1 | 1 | 0.34 |
| Yes | 354 (49.4) | 66 (18.6) | 1.16 (0.77, 1.75) | 1.22 (0.81, 1.83) | |
| **In past 4 weeks, ever no food to eat in the household because of lack of resources** | | | | | |
| No | 473 (66.0) | 72 (15.2) | 1 | 1 | 0.19 |
| Yes | 244 (34.0) | 48 (19.7) | 1.29 (0.77, 2.16) | 1.41 (0.84, 2.39) | |
| **Wealth quintiles according to household assets** | | | | | |
| 1 –Lowest | 134 (18.8) | 19 (14.2) | 1 | 1 | 0.07 |
| 2 | 156 (21.9) | 30 (19.2) | 1.62 (0.86, 3.07) | 1.62 (0.86, 3.07) | |
| 3 –Medium | 122 (17.1) | 16 (13.1) | 1.05 (0.43, 2.61) | 1.05 (0.43, 2.61) | |
| 4 | 144 (20.2) | 18 (12.5) | 0.94 (0.45, 1.97) | 0.94 (0.45, 1.97) | |
| 5—Highest | 158 (22.1) | 36 (22.8) | 1.84 (0.98, 3.47) | 1.84 (0.98, 3.47) | |
| **Lifetime number of sex partners** | | | | | |
| 1 | 198 (27.6) | 19 (9.6) | 1 | 1 | 0.18 |
| 2 to 4 | 308 (43.0) | 55 (17.9) | 1.93 (1.06, 3.49) | 1.94 (1.03, 3.63) | |
| 5 to 9 | 147 (20.5) | 31 (21.1) | 2.07 (1.02, 4.19) | 2.14 (1.00, 4.58) | |
| 10+ | 64 (8.9) | 15 (23.4) | 2.10 (0.83, 5.31) | 2.26 (0.82, 6.26) | |
| **Condom used at last sex** | | | | | |
| **No** | 336 (46.9) | 58 (17.3) | 1 | 1 | 0.55 |
| Yes | 380 (53.1) | 62 (16.3) | 0.99 (0.62, 1.57) | 0.87 (0.55, 1.38) | |
| **Had sex in the last 12 months** | | | | | |
| No | 230 (32.1) | 35 (15.2) | 1 | 1 | 0.93 |
| Yes | 486 (67.9) | 85 (17.5) | 1.00 (0.66, 1.52) | 0.98 (0.66, 1.47) | |
| **Number of sex partners in the past 12 months (N = 486)** | | | | | |
| 1 | 300 (61.7) | 37 (12.3) | 1 | 1 | <0.001 |
| 2 to 4 | 147 (30.3) | 38 (25.9) | 2.57 (1.68, 3.92) | 2.54 (1.59, 4.07) | |
| >5 | 39 (8.0) | 10 (25.6) | 1.98 (0.78, 5.02) | 2.17 (0.78, 6.04) | |

OR = Odds ratio;

[1]. Also adjusted for study arm considering evidence of an effect among older men; In adjusted analyses, further adjusted for: Educational attainment and wealth index; with the exception of wealth index and food availability (see Methods).

spaces" due to their focus on maternal and child health, with AGYW more likely to attend for maternal health services and therefore be offered STI, including HIV, testing services [21]. These findings reinforce the need to deliver SRH services that are accessible and acceptable to ABYM in order to ensure equitable access to SRH services and achieve the Sustainable Development Goal target 3.7: universal coverage of SRH services [22,23].

Our findings reinforce the limitations of syndromic STI screening [24]; 6% of the over 7000 AYP screened reported experiencing any STI symptoms. The most common symptom at first STI screen was genital itching, particularly among AGYW, who are more likely to experience asymptomatic STIs and/or STI symptoms that are non-specific [25]. To improve the diagnosis of STIs among all AYP, there is a need for delivery of improved point-of-care diagnostics through integrated, community-based services [26]. In Zimbabwe, 33% of individuals attending integrated community-based SRH services for youth accepted point-of-care STI testing for chlamydia and gonorrhea, among whom 17% tested positive for chlamydia and/or gonorrhea [27]. In South Africa, a community-based mobile clinic delivering PrEP to AGYW found that almost 50% of AGYW who initiated PrEP tested positive for chlamydia and/or gonorrhea [28]. Due to financial constraints, Yathu Yathu could not offer point-of-care STI testing services; only 6% of AYP screened experienced STI-like symptoms and, in the endline survey, only 7% of AYP who ever STI-tested reported a positive STI test. Future iterations of the Yathu Yathu model of static, community-based and integrated SRH services delivered to youth by youth should ensure STI point-of-care (or near point-of-care) testing is available on site, particularly as these tests become available [26], to reach this priority population. The availability of incentives likely contributed to the relatively high number of AYP who opted for syndromic screening at the Yathu Yathu hubs; future iterations of the model should continue to offer incentives, including products to enhance dental and menstrual health [29]. In other countries and with other populations, outreach STI testing has proven acceptable to populations vulnerable to STIs [9,30].

Alongside the integration of point-of-care STI testing within healthcare services for AYP, services for youth should provide accurate information on STIs. Although youth in southern African countries with a high burden of HIV are exposed to HIV-related messaging and have comprehensive HIV knowledge [31], less information is provided on curable STIs [27]. Our risk factor analysis found that higher educational attainment was associated with higher levels of ever STI testing, similar to literature on uptake of HIV testing [32,33]. Furthermore, the role of condoms as the only currently available multi-purpose technology to prevent STI, including HIV, and unintended pregnancies should not be forgotten [34]. Among AYP in our study, just over one-third of AGYW and half of ABYM reported condom use at last sex; among AGYW, sexual behaviors were not associated with ever testing for an STI. Although for some AYP, the decision to use condoms may be driven by fertility desires and/or pleasure, and, for AGYW, may also reflect a reduced ability to negotiate condom use [35], for some, use may be informed by limited knowledge of the prevalence and consequences of symptomatic and asymptomatic STIs.

As expected, ever STI testing was higher among AGYW who reported ever giving birth. However, only 40% of AGYW who had ever given birth reported ever testing for an STI. With the prevalence of syphilis estimated at 3.4% in southern Africa [36], there remains a need to improve STI screening in ANC settings. Our risk factor analysis also found an association, among AGYW, between no food in the house in the past four weeks and ever STI-testing; similarly, there was evidence that AGYW residing in households with relatively less wealth were more likely to have ever STI tested. In population-based surveys conducted in six African countries, severe food insecurity, defined as having no food in the house more than three-times in the past month, was associated with an increased risk of HIV infection [37]. This increased risk was due, in part, to increased reliance on selling sex [37]. Although a tenuous

link in our study, AGYW who experience food insecurity and have less household-level wealth may be more likely to engage in selling sex, to know their risk of, and therefore to test for, STIs. An analysis of a nationally representative survey of adults aged 15–44 in the USA demonstrated an association between food insecurity and STI risk indicators (including previous diagnosis of chlamydia or gonorrhea) [38]. Similarly, a survey of university students in South Africa found an association between food insecurity and engaging in transactional sex for "money" or "to meet basic needs" [39]. Addressing the growing epidemic of STIs requires not only provision of youth-friendly SRH services but broader recognition of and interventions to address the structural drivers of vulnerability to STIs.

This study is subject to limitations. Firstly, the data on history of STI testing are self-reported and are subject to error. However, most studies on HIV testing also rely on self-reported data, and so our study is not unique in this regard. Second, due to the cross-sectional nature of our data, in our risk factor analysis, we cannot be sure of the temporal relationship between the explanatory variables and ever STI testing. The timing of some of our explanatory variables may be more recent than ever testing, for example, food insecurity in the last four weeks. Nonetheless, our risk factor analysis is informative about the characteristics and behaviors of youth who have STI tested. Lastly, implementation of Yathu Yathu was disrupted by COVID-19 control measures. In the absence of these measures, uptake of STI testing services may have been higher. Despite limitations, our study includes a large number of AYP and provides rigorous and critical evidence regarding population-level STI testing, data that are currently lacking among this priority population. We consider our findings generalizable to other urban areas that similarly experience a high burden of HIV.

## Conclusions

Syndromic STI screening through community-based SRH services led by youth had little impact on self-reported uptake of STI testing services. Few AYP in our study had ever or recently tested for STIs. There was, however, evidence of an impact on ever STI testing among young men, who remain underserved by available healthcare services. With a growing burden of STIs globally and AYP's increased vulnerability to STIs, there is a need for population-level data on the burden of STIs among AYP to advocate for and inform interventions to reach AYP with STI services.

## Supporting information

**S1 Checklist. CONSORT checklist.**
(DOCX)

**S2 Checklist. Inclusivity in global research.**
(DOCX)

## Author Contributions

**Conceptualization:** Bernadette Hensen, Sarah Fidler, Richard Hayes, Musonda Simwinga, Helen Ayles.

**Data curation:** Bernadette Hensen, Lucheka Sigande, Sian Floyd.

**Formal analysis:** Bernadette Hensen, Ab Schaap, Sian Floyd, Richard Hayes.

**Funding acquisition:** Bernadette Hensen, Sian Floyd, Sarah Fidler, Richard Hayes, Musonda Simwinga, Helen Ayles.

**Investigation:** Bernadette Hensen, Mwelwa M. Phiri, Ab Schaap, Melvin Simuyaba, Richard Hayes, Helen Ayles.

**Methodology:** Bernadette Hensen, Mwelwa M. Phiri, Lucheka Sigande, Ab Schaap, Rosemary Zulu-Phiri, Sian Floyd, Sarah Fidler, Richard Hayes, Musonda Simwinga, Helen Ayles.

**Project administration:** Louis Mwape, Musonda Simwinga.

**Resources:** Mwelwa M. Phiri.

**Supervision:** Rosemary Zulu-Phiri, Louis Mwape, Musonda Simwinga, Helen Ayles.

**Writing – original draft:** Bernadette Hensen.

**Writing – review & editing:** Bernadette Hensen, Mwelwa M. Phiri, Sian Floyd, Sarah Fidler, Richard Hayes, Helen Ayles.

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
