## [Decision Letter · Decision Letter 0]

25 Oct 2023

PGPH-D-23-01825

Self-reported uptake of STI testing services among adolescents and young people aged 15-24 years: findings from the Yathu Yathu trial in Lusaka, Zambia

Dear Dr. Hensen,

Thank you for submitting your manuscript to PLOS Global Public Health. After careful consideration, we feel that it has merit but does not fully meet PLOS Global Public Health’s publication criteria as it currently stands. Therefore, we invite you to submit a revised version of the manuscript that addresses the points raised during the review process.

We look forward to receiving your revised manuscript.

Kind regards,

Joel Msafiri Francis, MD, MS, PhD

Academic Editor

Journal Requirements:

2. Please provide separate figure files in .tif or .eps format.

Additional Editor Comments (if provided):

Reviewers' comments:

Reviewer's Responses to Questions

**Comments to the Author**

1. Does this manuscript meet PLOS Global Public Health’s publication criteria? Is the manuscript technically sound, and do the data support the conclusions? The manuscript must describe methodologically and ethically rigorous research with conclusions that are appropriately drawn based on the data presented.

Reviewer #1: Yes

Reviewer #2: Yes

2. Has the statistical analysis been performed appropriately and rigorously?

Reviewer #1: Yes

Reviewer #2: No

3. Have the authors made all data underlying the findings in their manuscript fully available (please refer to the Data Availability Statement at the start of the manuscript PDF file)?

Reviewer #1: Yes

Reviewer #2: Yes

4. Is the manuscript presented in an intelligible fashion and written in standard English?

Reviewer #1: Yes

Reviewer #2: Yes

5. Review Comments to the Author

Reviewer #1: This is an important addition to the literature on ST testing for AYP in Sub-Saharan Africa, focused on self-reported uptake amongst AGYW and ABYM in Zambia as part of the Yathu Yathu trial. The findings show low impact on self-reported uptake of STI testing amongst AYP in general with some effect amongst ABYM aged between 20 and 24. The study, described in detail in a previous publication, was co-designed and conducted with the involvement of AYP. Relatively high numbers of AYP presenting for syndromic STI screening was likely a result of the incentivisation process through the PPC. This influence might have been considered further in the Discussion section, particularly when the authors argued for structural interventions to address drivers of vulnerability to STIs amongst AYP. Similarly, the association between household poverty and STI exposure amongst AGYW might have been strengthened by the inclusion of evidence for what the authors themselves describe as 'a tenuous link'. However, other than these very small points, the paper is very well written and the analysis rigorously conducted and described. Overall the paper is ready for publication and I would recommend its acceptance.

Reviewer #2: General comments

1. It is not clear what it is that was offered to the adolescents and young people in the control arm, besides the prevention points cards that seemed to be offered to all participants. A clear description of the services in the control arm, which the participants accessed, preferably as part of routine care, should be provided.

2. The authors indicate the following study outcomes: “the percentage of AYP self-reporting ever testing for an STI, other than HIV, among AYP reporting ever having had sex, and the percentage of AYP self-reporting recently (last 12 months) testing for an STI among AYP reporting sex in the last 12 months”. I have 2 comments here:

a) Ever testing for STI could mean any testing done before the Yathu Yathu services became available in August 2019. How did the authors ensure that any testing referred to here was a result of the Yathu Yathu intervention? I thought that some of the variables used to assess the association with ever STI testing should have included exposure to the Yathu Yathu intervention, if the purpose was to link any STI testing to the intervention effect.

b) The authors collected data from two sources: the process data from the prevention point cards and the survey. How were the two data sources linked to ensure that the data were referring to the same person? For instance, how were the data collected through PPCs linked to the survey data to indicate that those who reported recent STI testing (in the last 12 months) had had access to STI services via Yathu Yathu?

c) In the main results, a logistic regression analysis is conducted for ever-STI testing. But the authors mention two different outcomes, including STI testing in the last 12 months. However, I did not any regression analysis for recent STI testing which to me would have been a more close measure of the impact of the intervention than ever-testing.

3. Since the Yathu Yathu was implemented as a CRT, the authors should include a CONSORT statement (extension to cluster randomized trials) as required in the reporting of such trials.

4. In general, I did not see how the data from PPCs (as one of the two data sources) were used to inform the reporting of results obtained through the endline survey (the second data source). For instance, I think that, based on data from PPCs, some 6.2% (n=679) of the participants reported any STI symptoms. Then, based on the results from the endline survey, unless I missed it, some 6.6% (n=23) reported ever having any STI symptoms. So, how did the authors use data from the PPCs and that from the endline survey during the reporting on the impact of Yathu Yathu on STI screening and testing among AYP? I think a response to these questions can help to justify if the two data sources were necessary.

Specific comments

1. Were the study communities divided into 10 or 20 zones? In the abstract, reference is made to 10 zones. However, on page 5 (study location and population), a total of 20 zones is used. Please clarify.

2. In the introduction section, the authors make a profound case for increases in STIs among adolescents aged 10-14 years, despite a general decrease in STIs among adolescents and young people (AYP) aged 10-24 years. However, their study recruited those aged 15-24 years. Why didn’t enroll the population group that is most at risk?

3. How did the authors define ‘STI screening’? How was it done?

4. How soon did the AYP go for STI testing after being screened with STI symptoms? Did the authors see any differences between arms? Did the authors find out if the AYP who tested positive for STIs (4.3% for syphilis and 0.9% for gonorrhea) sought treatment?

5. In the results section, the authors write: “At their first STI symptom screen, 7.7% (n=563) of AYP reported experiencing any STI symptom…” I am not sure I understand the denominator for this percentage, given that there is already a 6.2% (n=679) of AYP reporting symptoms. To improve this statement, the authors may revise thus: ‘… At their first STI symptom screening, 7.7% (n=563) of AYP who xxxx, reported experiencing any STI symptom”. This can help to qualify the 7.7% as a percentage derived from a defined sub-population.

6. In the results section, I suggest that the authors begin by describing the population studied (whose characteristics are shared in Table 1). At the moment, the results section begins with “Syndromic STI screening at Yathu Yathu services” – before the readers get to know the characteristics of the population studied.

7. Given differences in sexual behavior between adolescent boys and adolescent girls, I would present Table 1 stratified by sex (across study arms) so that we can see these differences between adolescent boys and girls by study arm across the different background characteristics.

8. In table 1, the authors should describe what the “wealth quintile” levels 1-5 stand for. A descriptor should be included. Besides, the term “wealth quintile” is used for a 3-tier arrangement. I don’t think it would be appropriate to refer to a 5-tier level with a “quintile”.

9. On page 9, the authors write, “Overall, 1989 AYP participated in the endline survey, among whom 75.7% (n=1505) reported ever having had sex and 1501 had complete data on testing for STIs”. It would be nice if the authors included a CONSORT diagram that displays how the final numbers in each arm that were used in the analysis (1501) were derived.

10. On page 11, the authors write, “Overall, 23.3% (n=350/1501) of AYP who ever had sex reported ever testing for an STI, other than HIV, and 13.5% (n=148/1099) of AYP who reported sex in the last 12 months also reported testing for an STI in the last 12 months”. This statement mixes two different results (i.e. ever STI testing in those who had ever had sex separately from those who had sex in the past 12 months, yet those who had sex in the past 12 months are part of those who have ever had sex) making interpretation of the statement a little difficult. I suggest that the authors separate it into two separate sentences:

a) Overall, 23.3% (350/1501) who ever had sex reported ever testing for an STI other than HIV; of these, xxx% (n=xxx) were sexually active in the past 12 months of whom xxx% (xxx/yyy) reported that they tested for STIs in the past 12 months.

b) Of 1,099 AYP who reported sex in the past 12 months, xxx% (xxx/yyy) reported having any STI symptoms; of these, xxx% (xxx/yyy) reported that they tested for STIs in the past 12 months; or something in this regard.

11. Page 14 begins, “Among those who ever tested for an STI, 6.6% (n=23) reported ever being diagnosed with an STI; five individuals did not know which STI they tested positively for.” It is not clear to me – which denominator was used to derive the reported 6.6%. I suggest that whenever sub-group analyses are reported, the authors should describe the population sub-group referred to and provide a denominator to help in qualifying the percentage.

12. About Table 3:

a) How did the authors ensure that the ever-STI testing was associated with the Yathu Yathu intervention? Why didn’t the authors determine the factors associated with recent STI testing, which was more likely to be associated with the intervention, given the timing?

b) Why did the authors opt to use odds ratios when the prevalence of the outcome was higher than 10%?

c) It is not clear if the results presented have been adjusted for clustering effect and exposure to the intervention.

d) These comments also apply to Table 4.

6. PLOS authors have the option to publish the peer review history of their article (what does this mean?). If published, this will include your full peer review and any attached files.

**Do you want your identity to be public for this peer review?** For information about this choice, including consent withdrawal, please see our Privacy Policy.

Reviewer #1: No

Reviewer #2: No

---

## [Editor Report · Decision Letter 1]

19 Dec 2023

Self-reported uptake of STI testing services among adolescents and young people aged 15-24 years: findings from the Yathu Yathu trial in Lusaka, Zambia

PGPH-D-23-01825R1

Dear Dr. Hensen,

We are pleased to inform you that your manuscript 'Self-reported uptake of STI testing services among adolescents and young people aged 15-24 years: findings from the Yathu Yathu trial in Lusaka, Zambia' has been provisionally accepted for publication in PLOS Global Public Health.

Best regards,

Joel Msafiri Francis, MD, MS, PhD

Academic Editor